∂ | **Open Peer Review** | Host-Microbial Interactions | Research Article
# Metagenome-assembled genomes from a population-based cohort uncover novel gut species and within-species diversity, revealing prevalent disease associations

Kateryna Pantiukh,[1] Kertu Liis Krigul,[1] Oliver Aasmets,[1] Elin Org[1]

**ABSTRACT** Metagenomic profiling has advanced the understanding of microbe-host interactions. However, widely used read-based approaches are limited by incomplete reference databases and the inability to resolve strain-level variation. Here, we present a scalable, genome-resolved framework that integrates population-specific metagenome-assembled genomes (MAGs) to discover novel species, within-species diversity, and disease associations. From 1,878 deeply sequenced samples in the Estonian Microbiome Cohort (EstMB-deep), we reconstructed 84,762 MAGs representing 2,257 species, including 353 (15.6%) previously uncharacterized species reaching up to 30% relative abundances in some individuals. We integrated these MAGs with the Unified Human Gastrointestinal Genome collection to create an expanded reference (GUTrep), enabling profiling of 2,509 EstMB individuals and testing associations with 33 prevalent diseases. Of the 25 diseases with significant associations, 8 involved newly identified species, underscoring the value of population-specific MAGs. To quantify within-species diversity, we developed the genome unit number (GUN), a novel MAG-based metric that informed within-species analyses. Based on normalized GUN, we prioritized *Odoribacter splanchnicus,* a prevalent species with the lowest within-species heterogeneity, yielding sufficient power for a within-species association study. We identified two dominant genome units, GU-N1 and GU-N2, with distinct gene repertoires and divergent disease associations. Notably, GU-N1 was negatively associated with gastritis, duodenitis, and hypertensive heart disease, associations undetected at the species level. Our study expands the human gut reference landscape, demonstrates the importance of population-specific MAGs for uncovering novel microbial diversity, and reveals new disease associations at the within-species level obscured at higher taxonomic levels, highlighting the need for genome-resolved approaches in microbiome research.

**IMPORTANCE** Microbiome studies increasingly recognize that species-level profiles can mask critical within-species differences relevant to health and disease. However, our work shows that within-species diversity varies drastically across gut microbes, with some species exhibiting almost as many distinct within-species clusters as recovered genomes, making association studies at the within-species level essentially intractable. To address this, we introduce the genome unit number (GUN), a scalable metric for quantifying within-species structure. Using GUN, we demonstrate that only species with limited within-species diversity, such as *Odoribacter splanchnicus*, currently allow for robust within-species association testing. These findings emphasize the need to systematically evaluate species structure across the gut microbiome and call for the development of new computational and statistical approaches to enable meaningful within-species analyses in highly diverse species.

**Peer Reviewer** Luis Miguel Rodriguez-Rojas, Universität Innsbruck, Innsbruck, Austria

Address correspondence to Kateryna Pantiukh, pantiukh@ut.ee, or Elin Org, elin.org@ut.ee.

The authors declare no conflict of interest.

See the funding table on p. 15.

**KEYWORDS** gut microbiome, metagenome-assembled genomes, genome unit number, population microbiome, metagenomics, metagenome-wide association study, within-species diversity

The human gut microbiome exhibits remarkable diversity across individuals and populations, necessitating comprehensive global reference databases to enable accurate taxonomic and functional profiling of microbial communities. In recent years, considerable research effort has been directed toward establishing collections of global reference genomes of the human gut microbiome. Initially, the focus was on sequencing bacteria that could be isolated and cultured (1, 2). However, rapid technological advancements have facilitated the generation of vast amounts of metagenomic data and the development of techniques for assembling genomes from unculturable species, consequently improving reference databases. These metagenome-assembled genomes (MAGs) substantially expand the number of gut microbial species, as 81% of the species in the current version of the Unified Human Gastrointestinal Genome (UHGG) collection were identified by MAGs while having no corresponding representative in any human gut culture database (3). Moreover, MAG assembly enables genome-centric analyses, such as identifying strains of species present in a population and conducting strain-level association studies (4, 5). Therefore, MAGs enable us to significantly improve our understanding of the ecosystem under study.

The importance of MAG recovery is exemplified in population biobanks that include deeply phenotyped individuals and their microbiome samples. In this case, it becomes possible to identify correlations between known, newly reconstructed species and specific genome structure and various environmental, dietary, or health-related factors. In recent years, several population-based biobanks with metagenomic data sets have been established; for example, the Dutch Microbiome cohort from the Lifelines biobank (6), the Israeli Project 10K cohort (7, 8), the FinRisk cohort (9), and the Estonian Microbiome (EstMB) Cohort from the Estonian Biobank (EstBB) (10). Analyses of these data sets have demonstrated that gut microbiome composition is associated with a range of environmental and lifestyle factors, particularly diet and medication use (10, 11), and that variation in the microbiome is associated with several diseases, such as cardiovascular diseases (12, 13), mental health disorders (14), and cancers (15, 16). Furthermore, emerging evidence suggests that the gut microbiome has predictive power, as demonstrated in the context of incident heart failure (17). However, many of these studies still rely solely on reference databases. These databases may lack representatives for many uncultured or underrepresented population-specific microbial species, leading to incomplete or biased interpretations.

In the present study, we leveraged deep metagenomic sequencing of a population-based Estonian microbiome-deep (EstMB-deep) cohort to assemble a comprehensive collection of MAGs, substantially expanding the reference database of human gut microbes with hundreds of previously uncharacterized species. We integrated these population-specific MAGs with public reference data and conducted association analyses with 33 prevalent diseases. To systematically assess within-species diversity, we developed a novel genome unit number (GUN) metric. We demonstrated its utility by identifying specific disease associations at the within-species level in *Odoribacter splanchnicus* that were not apparent at the species level. Our findings demonstrate that genome-resolved microbiome profiling can uncover novel disease-linked microbial signatures that remain hidden using conventional approaches.

## RESULTS

### Study design and cohort overview

The study aimed to first recover MAGs from a deeply sequenced Estonian gut microbiome cohort (EstMB-deep, *N* = 1,878) and expand the reference database by combining these newly assembled genomes with the existing public Unified Human

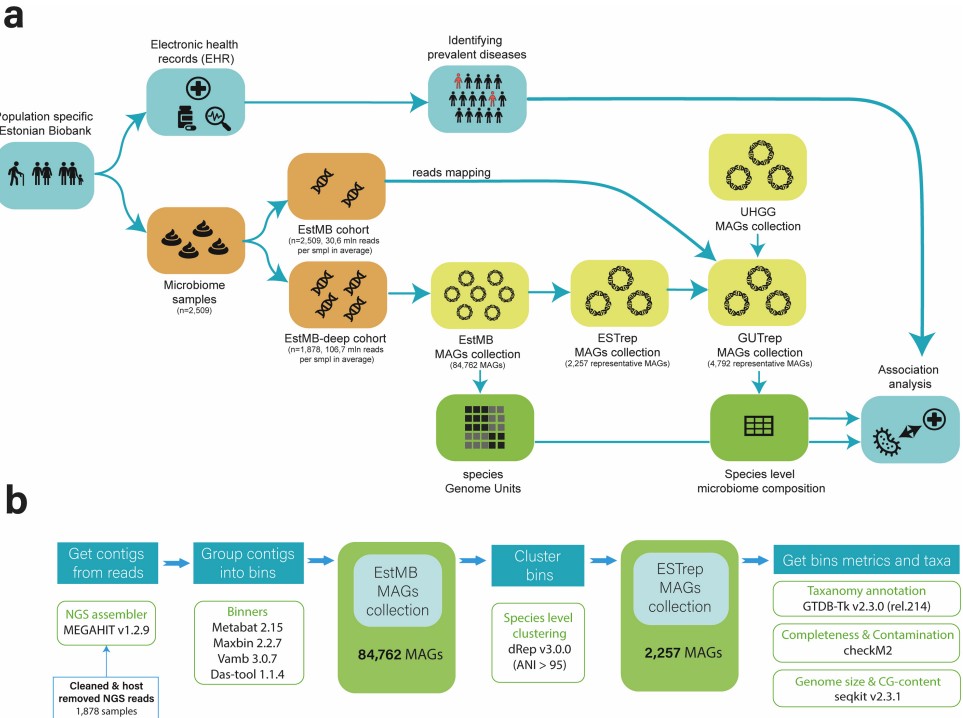

**FIG 1** Study overview and cohort description. (a) Study workflow overview. (b) Overview of the MAG recovery pipeline.

Gastrointestinal Genome collection and demonstrate the added value of genome-resolved, within-species analysis in identifying disease associations (3) (Fig. 1a).

The EstMB-deep subcohort used in the study is a subset of the volunteer-based Estonian Microbiome Cohort that is resequenced with much deeper coverage than the initial sample set (Fig. S1). In the EstMB-deep subcohort, each sample corresponds to a unique individual, with balanced distributions of age and gender. In brief, the EstMB cohort included 1,764 women (70.31%) and 745 men (29.69%), and the EstMB-deep subcohort consists of 1,308 women (69.65%) and 570 men (30.35%), with both cohorts representing individuals aged 23–89 (Fig. S2a). Compared to the EstMB average sequencing depth (30.63 ± 3.12 million reads per sample), EstMB-deep achieved over threefold higher coverage (106.70 ± 42.1 million; Fig. S2b). A detailed description of the EstMB, including omics and phenotypic data, is provided in Aasmets et al. (10).

## Creating a representative MAG pool of gut bacteria in the Estonian population

To characterize population-specific microbes and expand publicly available human gut microbiome databases with microbial genomes from the Estonian population, we performed *de novo* MAG reconstruction from all 1,878 samples in the EstMB-deep cohort. The MAG reconstruction pipeline is summarized in Fig. 1b. We successfully reconstructed 84,762 MAGs from EstMB-deep, with an average of 45.13 MAGs per sample. Among these, 42,049 (49.61%) were almost complete MAGs, i.e., MAGs with completeness > 90% and contamination < 5% according to CheckM. To describe the Estonian population species pool, we clustered all MAGs with dRep using a 95.0% average nucleotide identity (ANI) threshold, ensuring that the final clusters represent distinct species.

The species-level clustering procedure yielded 2,257 clusters (Table S1). For each cluster, the representative MAG was selected based on genome completeness, minimal contamination, strain heterogeneity, and $N_{50}$ (a parameter reflecting assembly fragmentation level). We refer to these 2,257 species representative MAGs as the "ESTrep" collection. The majority of ESTrep MAGs from the ESTrep collection (72.97%, *n*

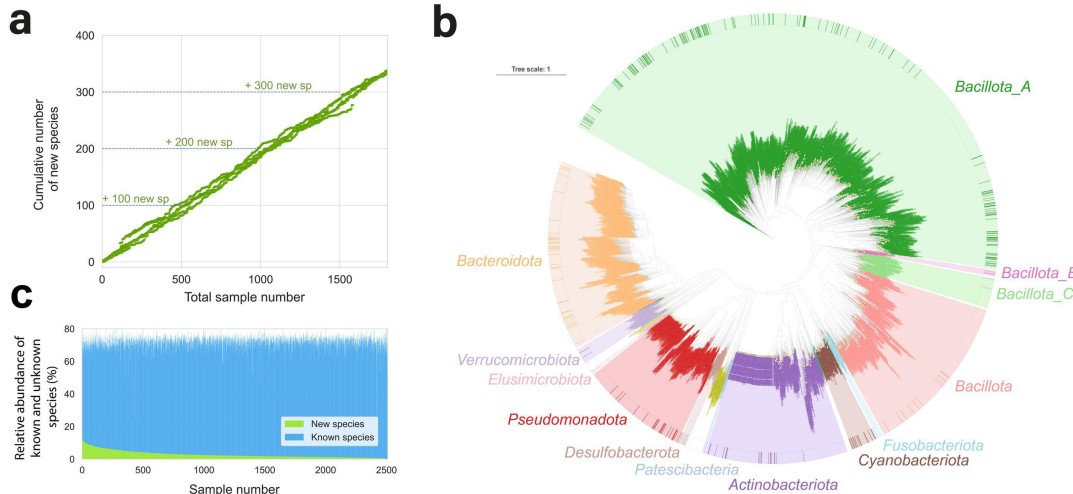

**FIG 2** Overview of novel species from the EstMB MAG collection. (a) The relationship between the number of samples analyzed and the cumulative number of novel species identified. (b) Phylogenetic tree of the ESTrep species. The inner circle displays a phylogenetic tree of species, with branches colored by phylum (according to the Genome Taxonomy Database [GTDB-Tk version 2.3.0]), the outer ring highlights novel species assembled in this study. (c) Relative abundances of known and novel species.

= 1,647 MAGs) were almost complete MAGs (MAGs >90% complete and <5% contaminated). Under the MIMAG standard (18), 475 representative MAGs (21.05%) qualified as high-quality, exhibiting >90% completeness, <5% contamination, and the presence of the 5S, 16S, and 23S rRNA genes together with at least 18 tRNAs. The remaining genomes included 1,714 MAGs (75.94%) of medium-quality (MQ) and 68 MAGs (3.01%) of low-quality (LQ) (Fig. S3).

## MAG assembly remains essential for detecting novel population-specific species

Next, we identified previously uncharacterized species within the ESTrep MAG collection. MAGs were categorized as novel species if their taxonomic classification at the species level or higher could not be assigned using the GTDB-Tk (19), a common approach for evaluating whether a newly reconstructed MAG represents a new species (20, 21). Of the 2,257 representative MAGs, 353 (15.64%) were classified as novel. Among these, 231 MAGs (65.44%) had >90% and <5% contamination, and 57 (16.15%) also contained rRNA and tRNA genes, meeting the MIMAG guidelines for high-quality MAGs (18). We observed a strong correlation between the number of novel species discovered and the number of samples analyzed ($R^2 = 0.97$). Specifically, for every 500 samples, approximately 102 novel species were identified (Fig. 2a). As we have not observed any indication of a plateau with the current sample size, we expect that analyzing more samples will reveal additional species.

Although Estonia is considered a Westernized population, novel species still make up a significant proportion of the microbiome community. These newly identified species were distributed across multiple phyla (Fig. 2b). On average, 2.82% of the total reads per sample were assigned to these novel species, even reaching a maximum relative abundance of 32.34% in some samples (Fig. 2c). Since these species are absent from public databases and may be population-specific, microbiome studies that rely solely on existing references may substantially underestimate microbial diversity.

## Integrating population-specific and global MAGs improves reference quality and uncovers assembly biases

As the success of metagenome assembly and genome reconstruction depends on multiple technical and analytical factors, we did not expect to recover all microbial

genomes present in the gut. Therefore, we constructed an integrated species-level reference by combining newly reconstructed MAGs from the Estonian population with publicly available human gut-associated species. This integrated reference, called the GUTrep collection (Fig. 1a), was generated by deduplicating the ESTrep MAG collection and UHGG MAG collection (22) at a 95.0% ANI threshold, retaining the highest-quality MAG for each species. When two MAGs from one species were present (one from ESTrep and one from UHGG), the highest-quality MAG was selected for the final collection. The final GUTrep database comprises 4,792 species, of which 3,285 (68.55%) originated from UHGG and 1,507 (31.45%) from ESTrep, thereby substantially improving the UHGG data set. Notably, the ESTrep contribution includes 353 novel species, 607 known species absent from UHGG, and 927 higher-quality MAGs already represented in UHGG.

To estimate microbiome composition across the EstMB data set ($n = 2,509$), we mapped all reads against the GUTrep collection. This approach, which does not require deep sequencing, identified 3,423 species in total. On average, each sample contained 292 species, whereas MAG assembly yielded an average of 45 MAGs per sample (Fig. 3a). The most prevalent (>95%) species detected by read mapping were all well-known gut microbes: *Phocaeicola dorei*, *Bacteroides* species (*Bacteroides uniformis*, *Bacteroides xylanisolvens*, and *Bacteroides ovatus*), *Faecalibacterium prausnitzii*, and *Odoribacter splanchnicus*. *P. dorei* and *B. uniformis* were also among the most abundant species, in addition to *Prevotella copri* (>2% on average) (Fig. S4). We observe that samples with more species detected by mapping also tended to have more MAGs recovered (Fig. S5). However, the number of assembled genomes per species did not clearly correlate with species prevalence or mean abundance (Fig. 3b; Table S2). Moreover, the difference between these values can range from minimal to substantial. For example, despite being one of the most prevalent species, *Bacteroides xylanisolvens* was detected in 97.13% of samples, with a mean relative abundance of 0.39%. However, only 18 MAGs were assembled for this species. This pattern, common among newly identified species, highlights that many species detected by mapping are represented only by a few MAGs, complicating genome-centric analysis (Fig. 3c; Table S3). Among 3,423 species detected, only 199 were represented by more than 100 assembled MAGs, and just 19 species had over 500 recovered MAGs (Fig. S6), illustrating the challenges of comprehensive genome reconstruction.

## Newly assembled species provide valuable input for association studies

Next, we utilized the comprehensive electronic health record (EHR) data from the Estonian population to perform a microbiome-wide association study (MWAS) of common diseases, using the population-based GUTrep reference. We included 33 prevalent diseases (≥100 cases each; Table S4), spanning various categories, such as the respiratory system (7 diseases), circulatory system (7 diseases), and digestive system (4 diseases) disorders. Associations between species abundance and diseases were assessed using linear regression models adjusted for BMI, gender, and age. To reduce multiple testing, we limited the analysis to species present in ≥1% of the samples, resulting in 1,595 species.

We identified 105 significant associations (Bonferroni-adjusted $P < 2.71 \times 10^{-5}$) between 96 bacterial species and 25 diseases (Tables S4 and S5). Notably, newly assembled species were associated with 8 out of the 33 diseases, including asthma, chronic ischemic heart disease, chronic rhinitis, nasopharyngitis and pharyngitis, female infertility, heart failure, hemorrhoids, iron deficiency anemia, and vitamin D deficiency. For example, one of the strongest associations was observed for chronic ischemic heart disease, involving a newly assembled species from the *Nanosynbacter* genus (species ID: H2144_Nanosynbacter_undS, adjusted $P = 3.13 \times 10^{-6}$) (Fig. 3d; Table S4). These findings emphasize the importance of population-specific reference databases for detecting disease-associated microbiome changes. However, further studies are needed to confirm whether these associations generalize beyond the Estonian cohort.

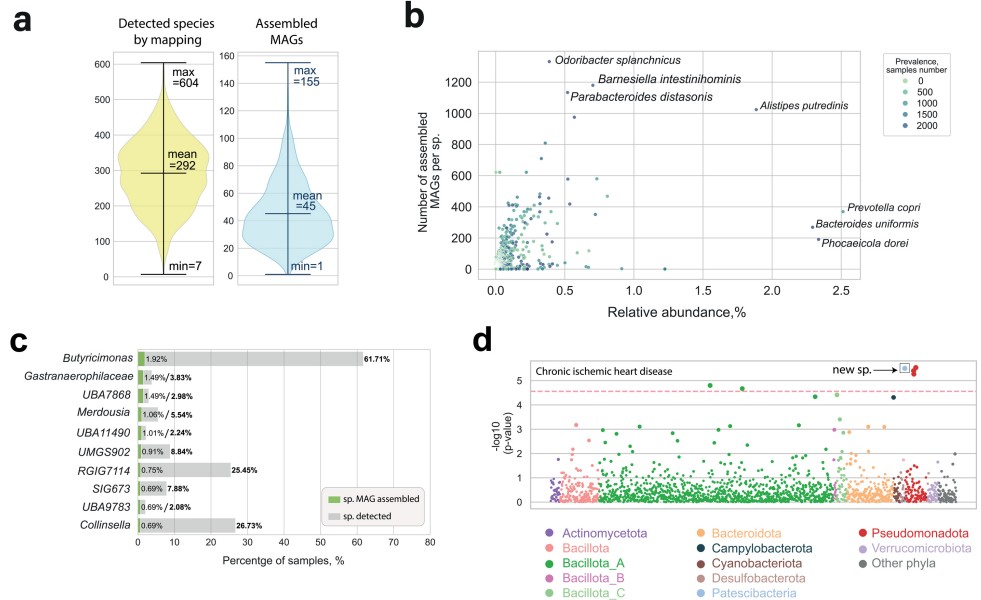

**FIG 3** Assembly–mapping comparison and microbiome-wide association study associations. (a) Average number of species detected by read mapping (yellow) versus number of recovered MAGs per sample (blue). (b) Relationship between species prevalence, mean relative abundance, and number of assembled MAGs per species. (c) Prevalence of the top 10 novel species with the highest number of recovered MAGs, comparing recovery by MAG assembly (green bars) and detection by read mapping (gray bars). (d) Metagenome-wide association results between GUTrep species abundances and chronic ischemic heart disease. Each data point corresponds to a single species, with vertical position reflecting the log-transformed *P*-value from linear regression; significant associations for newly reconstructed species are highlighted with a box.

## MAG data enable within-species genetic diversity analysis across species

Most large-scale microbiome association studies are conducted at the species level, although strain-level analysis is often recommended for understanding the functional insights (4, 23, 24). However, strain definitions vary substantially across studies, and many species do not exhibit genomic clustering patterns that provide sufficient sample sizes for robust within-species-level association testing. Metagenome assembly enables high-resolution reconstruction of species-level genomic diversity and allows the identification of taxa for which such analyses are feasible, thereby broadening the range of detectable biological and clinical associations. To capture this variation systematically and select species suitable for within-species-level MWAS, we introduce the genome unit number metric. GUN quantifies genomic diversity within a species by counting the number of highly similar genome units per species (ANI > 99%). We further define the normalized GUN (nGUN), which scales the GUN by the total number of MAGs recovered for the analyzed species, allowing comparison across species with differing MAG counts.

$$nGUN = (number\ of\ genome\ units/number\ of\ MAGs) \times 100\%$$

We focused on species with >10 reconstructed MAGs, yielding 376 species across diverse phyla. Notably, none of the newly identified species were included due to an insufficient number of MAGs.

The nGUN values varied widely, ranging from 0.4 to 100 (Fig. 4a; Table S6), indicating substantial differences in species population genetic diversity. *Odoribacter splanchnicus* exhibited the lowest nGUN (0.4), with one genome unit per ~250 MAGs, reflecting low diversity despite high prevalence. In contrast, *Prevotella copri* had one of the highest nGUN (94.0), consistent with its well-documented heterogeneity, where nearly every MAG represents a unique genome unit, making it difficult to conduct within-species

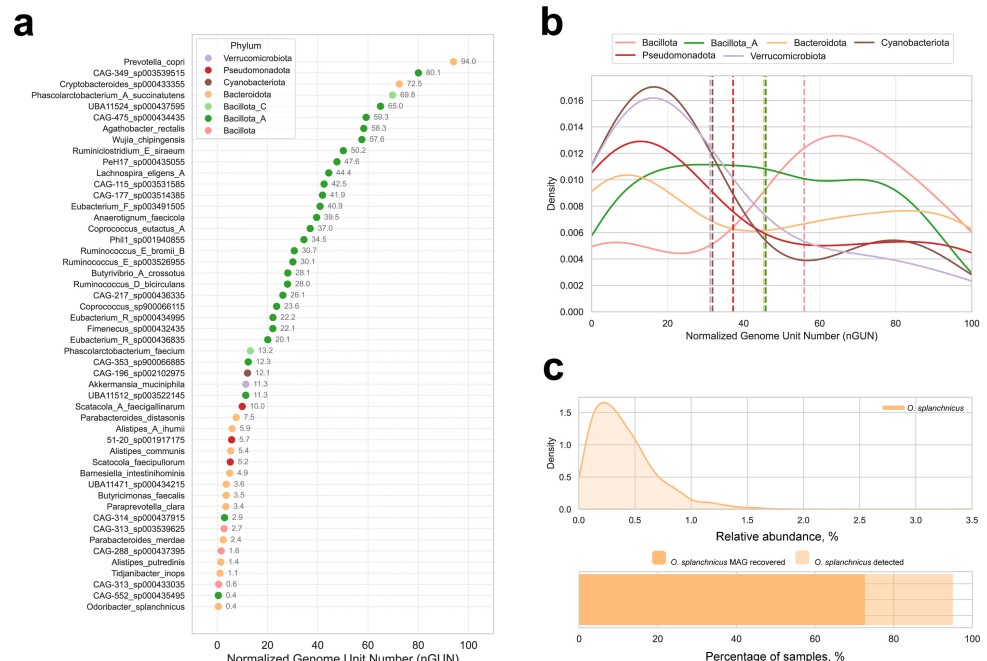

**FIG 4** Within-species diversity and within-species analysis of *Odoribacter splanchnicus*. (a) Normalized genome unit number values for the top 50 species with the highest number of MAGs. (b) Distribution of nGUN values across major gut bacteria phyla. (c) *Odoribacter splanchnicus* relative abundance, number of recovered MAGs, and prevalence across samples.

association analysis in the population. Interestingly, species from Alistipes_A genus appeared in both the lowest and highest nGUN groups.

We also examined nGUN distribution across six phyla with ≥10 species present in each: *Bacillota, Bacteroidota, Verrucomicrobiota,* Bacillota_A, *Pseudomonadota,* and *Cyanobacteriota,* all of which exhibited a broad range in nGUN distributions (Fig. 4b). *Bacillota* species tended to have higher nGUN values, indicating that this phylum generally tends to have a higher number of genome units per species. In contrast, *Verrucomicrobiota*, *Cyanobacteriota*, and *Pseudomonadota* exhibited lower nGUN values, suggesting that species in these phyla typically have fewer genome units per species. However, due to the small sample sizes in some phyla, further studies are needed to confirm whether these differences represent true phylum-level trends.

## nGUN-guided within-species analysis reveals phenotype associations missed at the species level

In order to demonstrate the value of nGUN-guided within-species metagenome-wide association analisys analysis, we selected *O. splanchnicus* due to its low nGUN (0.4) and high prevalence (detected in 96.14% of samples, assembled in 72.68%; Fig. 4c). Among its MAGs, we identified four distinct genome units, two of which were rare (found in 2 and 19 samples, respectively). Therefore, we focused on the two major genome units with high case numbers: genome unit N1 (GU-N1; *n* = 974 samples, original cluster ID: 1_2.3.4.6.9) and genome unit N2 (GU-N2; *n* = 335 samples, original cluster ID: 1_1) (Fig. 5a).

Logistic regression models adjusted for BMI, gender, and age were used to assess the association between the presence or absence of *O. splanchnicus* GU-N1 and GU-N2 and the same 33 diseases previously analyzed at the species-level MWAS. Our analysis identified a significant association between the presence of GU-N1 and two different diseases—gastritis and duodenitis, and hypertensive heart disease (Fig. 5b). The odds ratio for GU-N1 was less than 1 in both diseases (gastritis and duodenitis OR = 0.56, hypertensive heart disease OR = 0.63), indicating that its presence is associated with a

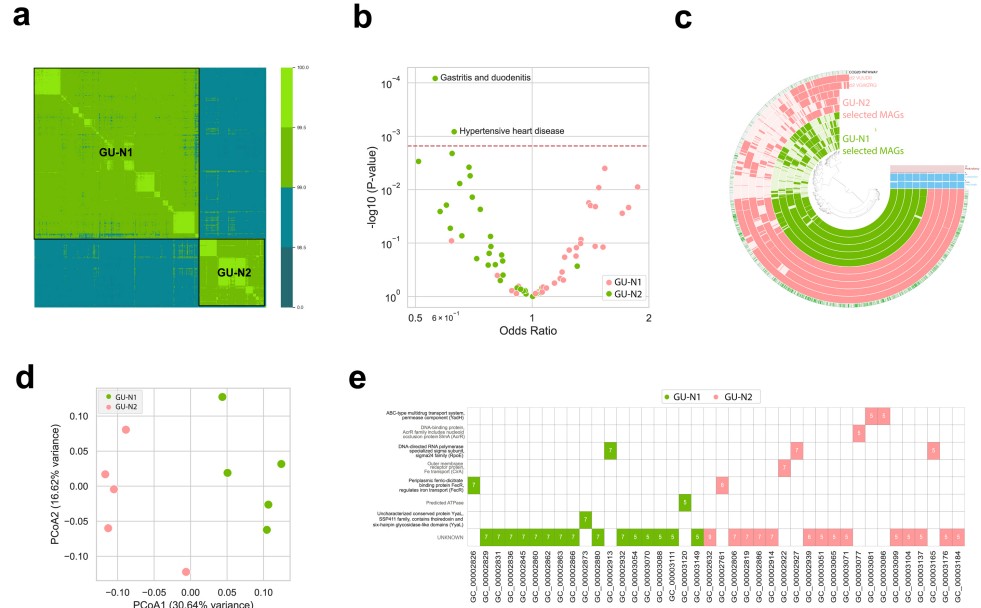

**FIG 5** Genomic structure and disease-associated variation across two major *O. splanchnicus* genome units. (a) Heatmap of average nucleotide identity values among *O. splanchnicus* MAGs, revealing two distinct genome units. (b) Volcano plot of associations between the two major *O. splanchnicus* genome units and 33 disease phenotypes. The red line indicates the Bonferroni-corrected significance threshold. (c) Pangenome analysis of five representative MAGs from each *O. splanchnicus* genome unit (GU-N1 and GU-N2). (d) Principal coordinate analysis (PCoA) of *O. splanchnicus* representative MAGs based on predicted gene cluster presence/absence profiles. (e) Gene clusters uniquely present in only one of the two major *O. splanchnicus* genome units.

reduced likelihood of having the disease. Notably, these associations were not detected at the species level, highlighting the added resolution of within-species-level analysis.

To explore functional differences, we performed a pangenome analysis of GU-N1 and GU-N2. We carried out principal coordinate analysis of predicted gene cluster presence/absence, which showed clear separation between the genome units (Fig. 5c). We identified that the two genome units formed distinct clusters, indicating clear genomic differentiation based on gene content (Fig. 5d), with 40 gene clusters unique to one of the two (Fig. 5e; Table S7). While most encoded hypothetical or uncharacterized proteins, some were annotated with putative functions based on the Clusters of Orthologous Genes (COG20) (25). GU-N2 harbored a broader repertoire of genes associated with stress response, iron acquisition, and antimicrobial resistance— traits consistent with enhanced survival in inflammatory gastrointestinal environments. These included elevated copy numbers of the extracytoplasmic stress sigma factor RpoE ($\sigma^E$), iron transport components FecR and CirA, and multidrug resistance elements such as AcrR and an ABC-type efflux pump (YadH). In contrast, GU-N1 was enriched for redox maintenance proteins such as YyaL/DsbD, suggesting a distinct strategy centered on oxidative stress mitigation.

## DISCUSSION

Our study presents a scalable, genome-resolution framework for population-scale microbiome analysis, enabling improved species and within-species characterization and discovery of disease associations. By expanding the gut microbial reference database with thousands of metagenome-assembled genomes, including novel bacterial species, as well as within-species genome units of known taxa, we address a major limitation in current reference data sets, which often underrepresent global microbiome diversity. We demonstrate that genome-resolution microbiome analysis, coupled with population-specific MAG catalogs, enables more comprehensive species- and within-species-level

association studies. Furthermore, we introduce the genome unit number, a novel quantitative metric of within-species genetic diversity, which we applied across 378 gut species to guide candidate selection for within-species-level analysis. Using this framework, we uncovered genome unit-specific disease signals for *O. splanchnicus* that were invisible at the species level and provided functional genomic insights that may explain these associations.

Read-based taxonomic profiling remains the most widely adopted approach to characterize microbial communities, particularly in association studies (26, 27). Alternatively, MAGs offer a culture-independent, reference-free approach to recover community structure (28). Despite deep sequencing (an average of 106 million read pairs per sample), metagenome assembly and binning reconstructed only ~45 genomes per sample on average, compared to 292 species detected by read mapping. This highlights that even abundant taxa are not always fully recoverable by MAG reconstruction, and *de novo* MAG assembly and binning alone fails to capture the full microbial diversity. This finding challenges the common assumption that high-abundance taxa can be reliably assembled given sufficient sequencing depth (29–31). We observed multiple cases where prevalent and relatively abundant species yielded few MAGs. For instance, *Bacteroides xylanisolvens,* present in 97.13% of samples at 0.39% mean abundance, yielded only 18 MAGs. Similar trends were observed for novel species, e.g., an undefined species from the *Butyricimonas* genus (ID: H0366) was assembled from just 36 samples but detected in >55% samples by mapping. These findings support earlier observations that low-abundance but genetically distinct species may assemble more readily than abundant, genetically diverse taxa (32) and underscore the need for a hybrid strategy combining MAG assembly and high-resolution read mapping against population-specific reference. Our GUTrep database exemplifies such a strategy, integrating local MAGs with the Unified Human Gastrointestinal Genome collection to improve reference coverage. Notably, 31% of dereplicated GUTrep species originated from our Estonian-specific MAGs, illustrating the added value of local assembly efforts.

A common argument against investing in resource-intensive *de novo* assembly is that the rapidly growing and regularly updated public gut genome catalogs increasingly capture known microbial diversity, suggesting that most gut species will soon be represented, and further assembly may become redundant. However, our findings challenge this assumption and support continued *de novo* assembly in new population studies. Early efforts reported high proportions of novel taxa, with 77% of MAGs classified as novel in Pasolli et al. (33) and 66% in Almeida et al. (22). More recent studies, such as Leviatan et al. (34), still report 310 novel species out of 3,594 assembled (8.6%). In our cohort, we recovered 353 novel species from 2,257 MAGs (15.6%), with ~102 additional novel species per 500 samples, and no indication of a discovery plateau. Moreover, we confirm a previously reported finding that many novel species assembled from a few samples were nonetheless widespread by mapping, suggesting that assembly remains essential even in well-characterized industrialized populations (32). These results reinforce the idea that local assembly efforts complement global references and remain critical for uncovering the full spectrum of microbial diversity.

Another advantage of *de novo* assembly is its ability to uncover within-species-level variation (4, 5). Strains of the same species can differ significantly in function and disease associations (35). As a classic example, well-known gut microbe *Escherichia coli* species includes strains that can be pathogenic (e.g., enterohemorrhagic O157:H7), probiotic (Nissle 1917), or commensal (K-12), and this demonstrates how it can be insufficient to study the microbe at the species level (36). While strain-level taxonomic profilers such as MetaPhlAn 4.0 (37) offer efficient resolution, they lack the genomic content necessary for detailed functional analyses, as they are usually based on marker gene analysis only. Other powerful tools, such as inStrain, operate at the whole-genome level and can report presence of different strains even among strains coexisting within the same sample (38).

In contrast, reconstructing MAGs directly from samples linked to host metadata allows for in-depth investigation of within-species genomic variation in relation to

specific phenotypes. Nevertheless, this approach is not feasible for all species in the population. Many taxa, especially newly discovered ones, are only recovered in a small number of samples, limiting their use in association analysis. In our data set, no newly identified species was assembled in more than 36 samples; the most prevalent was a novel species from the *Butyricimonas* genus (MAG ID: H0366). For species that are well represented in the MAG data set, high intraspecies genomic diversity can further complicate analysis. Thus, within-species genomic variability becomes a critical consideration when selecting candidate species for analysis on within-species level.

Our normalized genome unit number helps to assess whether within-species analysis is feasible by quantifying within-species diversity. High nGUN value of the species might indicate that each individual harbors a unique genome unit, complicating population-level associations. In this study, we analyzed within-species diversity across 378 gut species, expanding upon previous work that showed a strain richness of 92 gut species based on pure isolates from at least three different individuals (39). Our results significantly expand on this by including a broader range of species, each represented by more than 10 MAGs. Consistent with earlier findings, we observe substantial variability in within-species diversity across species (39). However, our data provide a more detailed and comprehensive picture due to the larger number of species. Among the 14 overlapping species between the two studies, some show similar patterns of genomic diversity—for example, *Odoribacter splanchnicus* and *Barnesiella intestinihominis* exhibit consistently low diversity, while others like *Bifidobacterium longum* remain highly diverse. Other overlapping examples, such as *Fusicatenibacter saccharivorans* and *Coprococcus eutactus*, display divergent diversity estimates, likely reflecting methodological differences (culture isolate versus metagenome-based) and highlighting the need for further comparative research. Highly diverse species such as *Prevotella copri,* well known from other studies (40, 41), are absent from the Chen-Liaw data set, but prominent in ours. Our larger data set also allowed the investigation of phylum-level patterns, suggesting that the common phyla, such as *Bacteroidota and Bacillota_A* species, tend to have higher nGUN values, while *Verrucomicrobiota, Cyanobacteriota,* and *Pseudomonadota* exhibit lower diversity. These phylum-level differences in within-species diversity suggest possible evolutionary or ecological constraints. However, further studies are needed, particularly for the less common phyla, to validate and understand the underlying mechanisms.

Understanding microbiome diversity at both species and within-species levels enhances resolution in metagenome-wide association studies. At the species level, we identified 96 bacterial species significantly associated with 25 common diseases, of which 8 diseases involved previously uncharacterized species, highlighting the limits of relying solely on global references. These differences may reflect population-specific variation driven by local differences in diet, genetics, and lifestyle (11). At the within-species level, we identified associations that were undetectable at the species level, such as the negative association of the *Odoribacter splanchnicus* genome unit GU-N1 with gastritis, duodenitis, and hypertensive heart disease. Previous research has shown that the abundance of the genus *Odoribacter* is negatively correlated with systolic blood pressure in overweight and obese pregnant women, suggesting that short-chain fatty acid-producing taxa may influence host blood pressure (42). Comparative genomic analysis of the MAGs from two *O. splanchnicus* genome units helped us to identify a set of gene clusters that differed between the two groups. These genomic features suggest that GU-N1 is functionally better adapted to conditions characteristic of gastritis and duodenitis, such as oxidative stress, nutrient limitation, and host antimicrobial pressure. Together, our findings highlight the value of genome-resolved metagenomic approaches in revealing disease-relevant microbial functions that would remain hidden in broader taxonomic analysis. While our study focused on a single, ethnically homogeneous Northern European population, several key findings, such as the detection of widespread yet previously uncharacterized species and species structures within common gut taxa, are likely to extend beyond the Estonian population. However, microbiome composition

is influenced by genetic, dietary, and environmental factors that vary between populations. Future studies in diverse cohorts will be essential to evaluate the generalizability of our results and validate the species- and within-species disease associations uncovered here. The genome-resolved analytical framework we present is scalable and readily applicable to other population-scale microbiome data sets, enabling cross-cohort comparison and discovery.

Our study also has some limitations that should be considered. First, the reliance on short-read sequencing may reduce assembly contiguity and within-species resolution compared to long-read approaches (43). Although long-read technologies offer higher genomic completeness, their current cost limits their use in large-scale population studies. A practical compromise could involve using short-read sequencing for most samples and applying long-read sequencing to key or novel taxa. Second, the observed associations are correlative, and further validation in longitudinal and experimental studies is needed to assess causality. Third, our association study on within-species level is focused on one species due to sample representation and analytical feasibility, and broader application across species remains a key next step. Finally, the genome unit number is influenced by multiple biological and technical factors that may bias its ability to accurately reflect within-species microdiversity. These include pangenome size, genome length, the propensity for horizontal gene transfer, and the level of homologous recombination. In addition, when a species is present within a sample as several highly similar genomic variants that may belong to different genome units, metagenomic assembly becomes more challenging. In such cases, MAGs may fail to be recovered or may represent composite genomes spanning multiple genome variants, which can lead to underestimation or conflation of true within-species diversity. Moreover, while functional differences between genome units of one species were identified, interpretation was limited by the prevalence of unannotated genes. Future improvements in genome annotation and cross-cohort replication will be essential to build on these findings. Despite these challenges, our findings demonstrate the value of population-scale metagenomics in uncovering novel microbial diversity and functional signatures relevant to human health.

## Conclusion

In conclusion, this study expands the human gut genomic reference, underscores the importance of population-specific MAGs in uncovering novel microbial diversity, and reveals within-species disease associations obscured at higher taxonomic levels, thereby highlighting the critical need for genome-resolved approaches in microbiome research.

## MATERIALS AND METHODS

### Estonian Microbiome Cohort description

The Estonian Microbiome Cohort was established in 2017, when stool, oral, and blood samples were collected from 2,509 EstBB participants (10). The EstBB is a volunteer-based population cohort initiated in 1999 that currently includes over 212,000 adults of European ancestry (≥18 years old) across Estonia (44). Extensive information is available for the EstMB participants, including data from self-reported questionnaires and EHRs (completed by medical professionals) covering diseases, medication use, and medical procedures both before and after sample collection. In addition to the questionnaire and EHR data, the participants' anthropometric measurements (e.g., height, weight, blood pressure, and waist and hip circumferences) were taken during a pre-registered visit upon delivering the stool sample. The Estonian Microbiome Deep cohort includes a subset of stool samples from the EstMB cohort that have been resequenced with over three times deeper coverage ($N = 1,878$).

## Microbiome sample collection and DNA extraction

The participants collected a fresh stool sample immediately after defecation with a sterile Pasteur pipette and placed it inside a polypropylene conical 15 mL tube. The participants were instructed to time their sample collection as close as possible to the visiting time in the study center. The samples were stored at −80°C until DNA extraction. The median time between sampling and arrival at the freezer in the core facility was 3 h 25 min (mean 4 h 34 min), and the transport time was not significantly associated with alpha diversity (Spearman correlation, $P$-value 0.949 for observed richness and 0.464 for Shannon index) or beta diversity (Bray–Curtis, $P$-value 0.061, $R^2 = 0.0005$). Microbial DNA extraction was performed after all samples were collected using a QIAamp DNA Stool Mini Kit (Qiagen, Germany). For the extraction, approximately 200 mg of stool was used as a starting material for the DNA extraction kit, according to the manufacturer's instructions. DNA was quantified from all samples using a Qubit 2.0 Fluorometer with a dsDNA Assay Kit (Thermo Fisher Scientific).

## Shotgun metagenomic sequencing

Sequencing for the main EstMB cohort was done using shotgun metagenomic paired-end sequencing on the Illumina NovaSeq 6000 platform and described in detail in reference 10. The EstMB-deep cohort samples were selected based on DNA quality and resequenced at higher depth using paired-end shotgun metagenomic sequencing on the MGISEQ-2000 platform. Sequencing reads' quality control (QC) was performed using FastQC (version 0.12.1) (45), and human reads were filtered using Bowtie2 (version 0.6.5) (46) against the GRCh38.p14 human genome reference. While following the QC, the EstMB cohort had an average of 30.63 ± 3.12 million reads per sample, whereas the EstMB-deep cohort resulted in 106.70 ± 42.1 million reads per sample, indicating over three times deeper sequencing coverage. Sequencing depth was evaluated with Nonpareil version 3.5.5 (47).

## EstMB MAG metagenome assembly and binning

The EstMB MAG collection refers to all MAGs recovered from the EstMB-deep cohort, which comprises 1,878 samples sequenced at deep coverage. Reads were assembled into contigs with MEGAHIT (version 1.2.9) (48). Binning was performed separately for each sample from the EstMB-deep cohort. Contigs were binned using binners: MetaBAT (version 2.15) (49), MaxBin (version 2.2.7) (50), and VAMB (version 3.0.7) (51), with further refining with DAS Tool (version 1.1.4) (52). MAGs resulting from this process form the EstMB MAGs collection. MAG quality, including completeness and contamination, was estimated using CheckM (version 2.3.1) (53).

## ESTrep MAG collection

ESTrep MAG collection refers to representative MAGs from the EstMB MAG collection described earlier. Representative MAGs were selected from the EstMB MAG collection by clustering MAGs from the EstMB MAG collection on the species level (ANI index = 95) with dRep (54). Taxonomy of all representative MAGs was assigned using GTDB-Tk (version 2.3.0) (19), a software toolkit for assigning objective taxonomic classifications to bacterial and archaeal genomes based on the Genome Taxonomy Database (GTDB) (19, 55). If a MAG could not be taxonomically classified at the species level or higher using GTDB-Tk, this indicates that the genome does not closely match any existing entries in the GTDB reference database. Therefore, it was treated as a novel species. This criterion is widely used in studies involving MAG assembly (20, 21). MAG completeness and contamination were estimated using CheckM (version 2.3.1) (53). Ribosomal RNA genes were identified with Barrnap version 0.8 (56), and tRNA genes were predicted using tRNAscan-SE version 2.0.0 (57).

MAGs were classified into three quality tiers. High-quality MAGs were defined as those with >90% completeness and <5% contamination and by the presence of ≥18 tRNA genes and a full complement of rRNA genes (5S, 16S, and 23S) (18). Medium-quality MAGs were defined as those with >50% completeness and <10% contamination. MAGs not meeting HQ or MQ thresholds were classified as LQ. Assembly statistics, including total assembly size, number of contigs, $N_{50}$, and GC content, were calculated using SeqKit (version 2.3.1) (58).

## Population-based reference GUTrep MAG collection

The GUTrep MAG collection is a non-redundant set of representative MAGs, created by combining MAGs from the current study (ESTrep MAGs) with those from the Unified Human Gastrointestinal Genome collection (22). This integrated reference includes both population-specific taxa identified in our cohort and globally distributed species that, while detected in our samples, could not be completely assembled but are present in public databases. To remove redundancy, MAGs from both collections were clustered at the species level using an average nucleotide identity threshold of 95% with dRep (54). For each species cluster containing MAGs from both sources, the higher-quality MAG—based on completeness, contamination, and assembly statistics—was retained as the representative.

## Species relative abundance and prevalence estimation

To evaluate species-level relative abundance and prevalence, we used all samples from the EstMB cohort, as it includes more samples than the EstMB-deep cohort. Deep sequencing is less critical for read profiling against an established reference database, whereas the larger sample size of the EstMB cohort is crucial for the subsequent association analyses. Reads were mapped against the GUTrep MAGs collection using CoverM (59) and aggregated into a relative abundance table with a custom Python script.

## Species-level association study

For the association study, we used species-level relative abundance data from the EstMB cohort as previously described. We tested associations between centered log-ratio (CLR)-transformed species abundances and participants' health status for common diseases in the Estonian population. We selected 33 diseases based on ICD10 codes from the electronic health records, each with at least 100 prevalent cases within the EstMB cohort. The remaining samples were considered as controls for each studied disease. From the 4,792 bacterial species in the GUTrep reference, we included 1,842 species with a prevalence >1% for the association analysis. Linear regression models, adjusted for BMI, gender, and age, were constructed to evaluate the association between the selected diseases and CLR-transformed species abundance. A stringent Bonferroni correction was applied to the significance level, adjusting for the number of analyzed species, resulting in a corrected alpha of $2.71 \times 10^{-5}$ (from an original alpha of 0.05).

## Genome unit number estimation

The GUN quantifies the number of genome units identified for a given species. Genome units are defined as clusters of genomes from the same species sharing an average nucleotide identity greater than 99% (analysis is restricted to species represented by more than 100 assembled MAGs). The nGUN represents the number of genome units per 100 MAGs and is calculated by dividing the species GUN by the total number of MAGs for that species and multiplying by 100 to express the value as a percentage.

The corresponding formula is shown below:

$$nGUN = \frac{\text{species GUN}}{\text{total number of MAGs}} \times 100\%$$

## Within-species association study

Candidate species for within-species association analysis were selected based on two criteria: (i) a high number of reconstructed MAGs per species and (ii) the lowest normalized GUN, indicating fewer within-species clusters per species. These criteria were established to ensure sufficiently large sample sizes for robust microbiome-wide association studies. Genome units were defined using dRep (54). Based on these criteria, *O. splanchnicus* was selected as the candidate species for within-species analysis. We examined the within-species population structure of *O. splanchnicus* and focused on two out of the five most prevalent identified *O. splanchnicus* genome units. The remaining three genome units were excluded due to their presence in only a small subset of samples. For association analyses, we used presence/absence data from the two selected genome units. These genome units were designated as GU-N1 ($n = 974$; original cluster ID: 1_2.3.4.6.9) and GU-N2 ($n = 335$; original cluster ID: 1_1). Logistic regression analyses adjusted for sex, age, and BMI were performed for the same 33 diseases previously examined at the species level. To account for multiple testing, a stringent Bonferroni correction was applied, resulting in a corrected significance threshold of $\alpha = 1.5 \times 10^{-3}$ (original $\alpha = 0.05$).

For species cluster structure visualization, we used ANIclustermap (60). Pangenome analysis and pangenome visualization were performed using the Anvi'o workflow with standard parameters (61).

## ACKNOWLEDGMENTS

We express our gratitude to all individuals who made valuable contributions to the Estonian Microbiome cohort, as well as to those who developed the software and databases utilized in this study. We thank Mait Metspalu, Andres Metspalu, Lili Milani, and Tõnu Esko from the Estonian Biobank research team for the Estonian Biobank health data collection. Data analysis was carried out in part in the High-Performance Computing Centre of the University of Tartu, and we thank the HPC Support Team of the Institute of Computer Science at the University of Tartu for delivering exceptional service and assistance in installing the necessary programs on the cluster. This work was written at writing retreats and writing days organized by the Institute of Genomics, University of Tartu. We also thank Million Metagenomes of Human Project (MMHP) for providing the sequencing for the EstMB-deep cohort.

This work was funded by the Estonian Research Council grant (PRG1414 to E.O.) and an EMBO Installation grant (No. 3573 to E.O.). EstMB sample collection was supported by the Estonian Center of Genomics/Roadmap II project No. 16-0125.

K.P., K.L.K., and E.O. conceptualized the study; K.P. and O.A. designed the methodology; K.P. analyzed the data, visualized the study, and prepared the first draft of the manuscript, which all authors reviewed and edited. All authors agreed to submit the manuscript, read and approved the final draft, and assumed full responsibility for its content, including the accuracy of the data.

## AUTHOR AFFILIATION

[1]Institute of Genomics, Estonian Genome Centre, University of Tartu, Tartu, Estonia

## AUTHOR ORCIDs

Kateryna Pantiukh  http://orcid.org/0000-0002-2595-0673
Kertu Liis Krigul  http://orcid.org/0000-0002-4195-7357
Oliver Aasmets  https://orcid.org/0009-0001-9872-6031

Elin Org 🔟 http://orcid.org/0000-0001-8688-9717

## FUNDING

| Funder | Grant(s) | Author(s) |
|---|---|---|
| Estonian Research Council Grant | PRG1414 | Elin Org |
| EMBO Installation grant | 3573 | Elin Org |
| Estonian Center of Genomics/Roadmap II | 16-0125 | Elin Org |

## AUTHOR CONTRIBUTIONS

Kateryna Pantiukh, Conceptualization, Data curation, Formal analysis, Investigation, Methodology, Resources, Software, Visualization, Writing – original draft | Kertu Liis Krigul, Conceptualization, Validation, Writing – review and editing | Oliver Aasmets, Methodology, Writing – review and editing | Elin Org, Conceptualization, Supervision, Writing – review and editing

## DATA AVAILABILITY

The source code for the analyses is available at GitHub: https://github.com/Chartiza/EstMB_MAGs_db_paper. Representative MAGs from the EstMB-deep cohort samples have been deposited in the European Nucleotide Archive under study accession PRJEB76860. The phenotype data contain sensitive information from healthcare registers, and they are available under restricted access through the Estonian biobank upon submission of a research plan and signing a data transfer agreement. All data access to the Estonian Biobank must follow the informed consent regulations of the Estonian Committee on Bioethics and Human Research, which are clearly described in the Data Access section at https://genomics.ut.ee/en/content/estonian-biobank. A preliminary request for raw metagenome and phenotype data must first be submitted via the email address releases@ut.ee.

## ADDITIONAL FILES

The following material is available online.

### Supplemental Material

**Supplemental Figures (mSystems00114-26-s0001.docx).** Fig. S1 to S6.
**Supplemental Tables (mSystems00114-26-s0002.xlsx).** Tables S1 to S7.

### Open Peer Review

**PEER REVIEW HISTORY (review-history.pdf).** An accounting of the reviewer comments and feedback.

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
