## [Reviewer comments · mSystems]

Metagenome-assembled genomes from a population-based cohort uncover novel gut species and within-species diversity, revealing prevalent disease associations

Kateryna Pantiukh, Kertu Krigul, Oliver Aasmets, and Elin Org

Corresponding Author(s): Kateryna Pantiukh, Tartu Ulikool

Review Timeline:

Submission Date:

January 31, 2026

Accepted:

February 10, 2026

Editor: Alejandro Reyes Munoz

Reviewer(s): Disclosure of reviewer identity is with reference to reviewer comments included in decision letter(s). The following individuals involved in review of your submission have agreed to reveal their identity: Luis Miguel Rodriguez-Rojas (Reviewer #2)

Transaction Report:

DOI: <https://doi.org/10.1128/msystems.00114-26>

Re: mSystems00114-26 (**Metagenome-assembled genomes from a population-based cohort uncover novel gut species and within-species diversity, revealing prevalent disease associations**)

Dear Dr. Kateryna Pantiukh:

Although the reviewers have agreed on the suitability of the manuscript for publication, reviewer 1 still has some minor comments that should be attended while making editorial changes

Your manuscript has been accepted, and I am forwarding it to the ASM production staff for publication. Your paper will first be checked to make sure all elements meet the technical requirements. ASM staff will contact you if anything needs to be revised before copyediting and production can begin. Otherwise, you will be notified when your proofs are ready to be viewed.

Sincerely,
Alejandro Reyes Munoz
Editor
mSystems

Reviewer #1 (Comments for the Author):

The authors have done an excellent job revising the manuscript and addressing my previous comments. The revised version is substantially improved in clarity and presentation, and I support acceptance. I only note a few minor editorial/consistency corrections to implement in the final version:

1. When you use "metagenome-assembled" as an adjective, please hyphenate it. Also, define acronyms only once: use MAGs after the first definition (avoid repeating "metagenome-assembled genomes"), and similarly define Genome Unit Number (GUN) once and then use GUN consistently.
2. Please replace "sub-species" terminology with "within-species" terminology throughout.
3. Please hyphenate quality tiers when used as modifiers (e.g., high-quality MAGs, medium-quality MAGs, low-quality MAGs) (L128, L131).
4. Please standardize number formatting with thousands separators where appropriate (e.g., 1,878 samples; L115, L467).
5. Correct decimal formatting to English style (decimal points, not commas): 75.94% and 3.01% (L130).

6. Resolve the internal inconsistency in the HQ MAG definition regarding tRNA requirement: {greater than or equal to}21 tRNA genes (L515) vs {greater than or equal to}18 tRNAs (L130)-please align the threshold across Methods and Results.
6.Minor edits: "FIG4." → "FIG 4." (L261) and "gut bacteria phyla" → "gut bacterial phyla" (L264).

Once these small edits are made, the manuscript will be ready for publication.

Reviewer #2 (Comments for the Author):

The authors have satisfactorily addressed all major issues and significantly improved the manuscript.